# Adolescent Sexual and Reproductive Health Care Service Availability and Delivery in Public Health Facilities of Plateau State Nigeria

**DOI:** 10.3390/ijerph18041369

**Published:** 2021-02-03

**Authors:** Esther Awazzi Envuladu, Karlijn Massar, John de Wit

**Affiliations:** 1Department of Community Medicine, College of Health Sciences, University of Jos, Jos P.M.B 2084, Nigeria; 2Department of Work & Social Psychology, Maastricht University, P.O. Box 616, 6200 MD Maastricht, The Netherlands; karlijn.massar@maastrichtuniversity.nl; 3Department of Interdisciplinary Social Science, Utrecht University, P.O. Box 80140, 3508 TC Utrecht, The Netherlands; j.dewit@uu.nl

**Keywords:** adolescent, sexual and reproductive health, availability, service delivery, Nigeria

## Abstract

To assess the availability, accessibility, appropriateness and quality of adolescent sexual and reproductive health (ASRH) services in primary health care (PHC) facilities in Plateau State, Nigeria, a cross-sectional study was conducted in 230 PHC facilities across the three senatorial zones of Plateau state. Primary data were obtained through face-to-face interviews with heads of facilities from December 2018 to May 2019. An adapted questionnaire from the World Health Organization (WHO) was used, covering five domains, to ascertain the extent that ASRH services were available and provided. Very few PHC facilities in the state had space (1.3%) and equipment (12.2%) for ASRH services. The proportion of PHC facilities offering counselling on sexuality was 11.3%, counselling on safe sex was 17%, counselling on contraception was 11.3% and management of gender-based violence was 3%. Most facilities were not operating at convenient times for adolescents. Only 2.6% PHC facilities had posters targeted at ASRH and just 7% of the PHCs had staff trained on ASRH. These findings underscore that the majority of PHC facilities surveyed in Plateau State, Nigeria, lacked dedicated space, basic equipment, and essential sexual and reproductive health care services for ASRH, which in turn negatively affect general public health and specifically, maternal health indices in Nigeria. Structural changes, including implementation of policy and adequate additional training of healthcare workers, are necessary to effectively promote ASRH.

## 1. Introduction

Adolescents worldwide, including in sub-Saharan Africa, are known to engage in sexual behaviours that place them at risk of adverse sexual and reproductive health outcomes, such as condomless sex and multiple sexual partners [1]. Despite a declining trend in HIV incidence globally, the HIV infection rate remains high among adolescents in sub-Saharan Africa, including Nigeria [1,2]. Some studies have reported an HIV prevalence of 14–17% among adolescents in Nigeria, and the prevalence of other sexually transmissible infections (STIs) among adolescents in Nigeria has been found to range from 29% to 48.8% in different parts of the country [3,4,5].

In addition to increased risk of infection with HIV and other STIs, condomless sex is also related to the high rates of teenage pregnancy that are reported in most sub-Saharan African countries [6,7,8]. A systematic review of evidence from 52 African countries reported an overall adolescent pregnancy rate of 18.8% in Africa and a pregnancy rate of 19.3% in sub-Saharan African countries [9]. This is similar to the pregnancy rate of 17% reported among adolescent girls in Plateau state, Nigeria, which is the region of focus of the current research [10]. In most instances, adolescent pregnancies in Nigeria end in termination, despite the illegality of abortion [11]. Nigeria records a yearly rate of 25 abortions per 1000 women [12,13], and a study conducted in the southern part of the country reported that 32% of these abortions are illegal, and often unsafe abortions occur among adolescent girls [13]. There is also evidence that some of these abortions are carried out by the adolescent girls themselves, using various dangerous local concoctions to terminate pregnancy [13,14]. In most cases, these girls end up with serious complications and some die in the process, adding to the burden of maternal mortality in Nigeria [15].

Furthermore, teenage pregnancies that are not terminated are associated with maternal reproductive risks for the teenage mother, as they are at higher risk of pre-eclampsia and eclampsia, antepartum haemorrhage and feto-pelvic disproportion, as well as obstructed labour and its sequelae, notably genital fistulae. Teenage pregnancy also contributes significantly to maternal mortality rates [16]. Furthermore, there are also potential social and psychological consequences for these girls. For example, teenage pregnancies in low and middle income countries can, and often do, end in child marriage, whereby girls are forced to marry the men responsible for the pregnancy [17]. Indeed, unintended pregnancy has been identified as a main reason for child marriage in Nigeria [18], and most married adolescent girls drop out of school, thereby becoming educationally and economically disadvantaged [19].

There is thus a clear need to reduce the risk of HIV and STIs as well as teenage pregnancy among adolescents in Nigeria. The World Health Organization (WHO) notes that the availability and access to quality sexual and reproductive health (SRH) information and services for adolescents is essential to reduce STI/HIV infections and teenage pregnancies [20]. The provision of these services therefore seems essential for Nigerian adolescents as well. However, although family planning and maternal health care services are provided in health facilities across Nigeria, these are often for adults, and not specific to adolescents’ needs [21,22]. Indeed, this lack of appropriate sexual and reproductive health care for adolescents is rapidly emerging as a public health concern in Nigeria, especially given the serious sexual and reproductive health risks adolescents face [22,23].

To meet the WHO recommendations, the Nigerian government introduced a policy about ten years ago to address the lack of adolescent-friendly sexual and reproductive health (ASRH) services in health facilities, especially in primary health care facilities, which are the first point of call and the nearest health facilities in communities [24]. The aim of this policy is to integrate ASRH services into primary health care facilities, to ensure the availability, accessibility, appropriateness and quality of ASRH in Nigeria. However, despite the existence of this policy, it appears these services are not yet provided as intended. Notably, the findings from an earlier study we conducted in Plateau state, Nigeria [25], showed that adolescents experienced various SRH challenges, including poor knowledge of SRH and poor health-seeking behaviour. Furthermore, the majority of adolescents in that study indicated they did not seek SRH services in health facilities, due to the absence of the services they required, an experienced lack of confidentiality or negative attitudes of health care providers [25].

Extending our earlier research on the sexual behaviour, SRH challenges and factors influencing the health seeking behaviour of adolescents, the objective of this study is to contribute to an assessment of the state of implementation of the national reproductive health policy for adolescents in Nigeria. Specifically, the aim was to assess the current provision of comprehensive ASRH services in primary health care facilities in Plateau State, Nigeria, and to compare if there are any differences across the senatorial zones related to the variation in the settings. Notably, the northern senatorial zone is an urban zone, the central zone is semi-urban and the southern zone is rural. To assess the ASRH services, we drew on the WHO guidelines for youth friendly health care services [26] that underscore the importance of ascertaining the extent to which the ASRH services are available, accessible, appropriate and of sufficient quality as determinants to the effective utilization of the services by adolescents in Plateau State.

## 2. Methods

### 2.1. Study Setting and Sampling

This study was conducted in 230 public primary health care facilities (PHC) in six Local Government Areas (LGA) in Plateau State, Nigeria. Plateau State is located in north-central Nigeria, and shares boundaries with Kaduna State (northwest), Bauchi State (northeast), Nasarawa State (southwest) and Taraba State (southeast). Plateau State has an estimated population of 3,206,531 (1,598,998 males and 1,607,533 females), with a growth rate of 2.8%; 32% of the population are adolescents [27]. There are 17 Local Government Areas across three senatorial zones. The northern zone has six LGAs, the central zone has five LGAs and the southern zone consists of six LGAs. 

We selected health facilities in two stages: First, two LGAs were selected from each of the three senatorial zones, using a simple random sampling technique. Second, we identified and included all the PHCs that provide services in the selected LGAs, while excluding PHCs that no longer provide services. The Cochran formula (n_0_ = Z^2^pq/e^2^) was used to calculate the sample size of health facilities to be included in the study [28]. The margin of error (e) for the sample size calculation was set at 5%, and the proportion of primary health facilities in each zone providing ASRH services (p; q = 1 − p) was set at 50%, equivalent to the national policy target. After applying the correction for smaller population sizes (nf = n_0_/1 + (n_0_ − 1)/N), the estimated required sample size was 217 facilities but eventually assessed 230 facilities.

### 2.2. Data Collection

A cross-sectional survey was conducted in all the included PHCs in the six selected LGAs across the three senatorial zones of Plateau state. Each participating PHC was visited by the research team, which consisted of the first author and ten research assistants who administered the survey questionnaires to the heads of the facilities through a face to face interview. Seven of the research assistants were resident doctors and three were community health workers. Team members were all trained before data collection commenced, and could speak English, Hausa and the common dialect in the study areas. A mapping of the number, the locations and distance between the PHC in each LGA was done to enable smooth logistics, considering that some PHCs are more than 5 km from the main access road. 

In addition to basic information, such as senatorial zone and LGA, the names and location of the health facilities were recorded. Further, we assessed the availability of space and equipment for ASRH, availability of specific ASRH services, accessibility of ASRH services, appropriateness of ASRH services and quality of ASRH services. The assessment of ASRH was based on the WHO guidelines for youth friendly health service [24]. The survey questionnaire was pre-tested in Jos South LGA, an LGA that was not included in the selected 6 LGAs in which study data were collected. 

Several questions were asked to assess each of the factors, as detailed below. The responses to each question were dichotomized, whereby affirmative responses (e.g., item is present, available, provided, carried out) were scored 1, and negating responses (e.g., item is not present, not available, not provided, not carried out) were scored 0. 

Availability of ASRH space and equipment was assessed with five items regarding the availability of a dedicated ASRH waiting area, a dedicated space for ASRH consultation, a dedicated area for ASRH counselling, a dedicated ASRH examination room and ASRH specific equipment (e.g., appropriate size speculum).

Availability of ASRH services was assessed with nine items regarding the provision of counselling on sexuality, prevention of pregnancy/contraception, safe sex/STI prevention and gender-based violence (GBV); as well as the management of GBV, voluntary counselling and testing (VCT) for HIV; and post-abortion care.

Accessibility of ASRH services was assessed with four items regarding distance of the facility from the main road, distance from places in the locality where adolescents gather, distance from school in the locality and opening/closing hours of the PHCs.

Appropriateness of ASRH services was assessed based on the availability of specific clinic hours for adolescent consultations, availability of posters and other ASRH educational materials, the availability of dedicated ASRH peer education staff, the provision of outreach services for adolescents, and the availability of ASRH services without parental consent.

Quality of ASRH services was assessed with three items regarding the availability of guidelines for provision off ASRH services, whether staff were trained on ASRH and the availability of referrals to specialized services if necessary.

### 2.3. Data Analysis

Data were analysed using SPSS version 23 (IBM Corp, Armonk, NY) For each of the three senatorial zones, we first computed the proportion of PHC responding affirmatively to each of the items pertaining the various domains of the adapted WHO guideline for ASRH (i.e., availability of space and equipment, availability of ASRH services, accessibility of ASRH services, appropriateness of ASRH services and quality of ASRH services). For each domain, we also established if a PHC responded affirmatively to all items, reflecting that a PHC met all criteria for that domain. We also compared the proportions of PHCs across the three senatorial zones that responded affirmatively to specific items per domain, using chi-square tests or Fisher’s exact tests, as appropriate. Furthermore, for each of the domains we compared the observed proportions of PHC per senatorial zone that met all criteria to the target of the National Sexual and Reproductive Health Policy stipulating that at least 50% of PHC should provide comprehensive ASRH services. 

### 2.4. Ethical Considerations

Ethical approval was obtained from the Jos University Teaching Hospital Human Research and Ethics Committee before the commencement of the study (reference number JUTH/DCS/ADM/127/XXVIII/1187). Permission to undertake the research was also obtained from the Ministry of Health, as well as from the chair persons of all LGAs. At each PHC facility, written informed consent was obtained from the officers who completed the questionnaires

## 3. Results

### 3.1. Overall Adolescent Friendliness of SRH Services

The findings from this study show that the state did not met the criteria for adolescent friendly SRH service as stated in the national policy. As can be seen in Table 1, all domains of ASRH received a poor score, indicating that, across facilities, substantially less services than recommended in the policy are being provided. For example, in the northern zone only 17.8% of PHCs had sufficient availability of ASRH services, and in the central zone only 13.5% of the facilities had ASRH services. In the southern zone, which had the highest score, only 22.8% facilities provided the recommended level of care. The differences in scores across all the domains was hover not statistically significant (*p* = 0.630).

### 3.2. Availability of ASRH Space and Equipment 

With regards to the availability of dedicated space and equipment for ASRH, the results shown in Table 2 highlight that very few PHC facilities in the state had a dedicated waiting area or consultation space for ASRH. Furthermore, only about one in ten facilities had a specific counselling area, only just over one in ten facilities had a dedicated examination room for ASRH and equally just over one in ten facilities had dedicated equipment for ASRH. There were no significant differences across the senatorial zones in terms of the proportion of facilities that had dedicated space for ASRH service delivery, a dedicated waiting area for ASRH or dedicated ASRH equipment. There were however significant difference between senatorial zones with respect to the availability of a dedicated ASRH counselling area and examination room. The southern zone had a significantly higher number of facilities with a dedicated counselling area than other zones, while facilities in the central zone were significantly less likely to have a dedicated ASRH examination room. 

### 3.3. Availability of ASRH Services 

As shown in Table 3, all availability of all assessed services was poor across the three senatorial zones. There were significant differences between the senatorial zones in the proportions of PHC that provided counselling on safe sex and post-abortion care. Safe sex counselling and post-abortion care were least likely to be provided in the central zone, and most likely to be provided in the southern zone. Provision was intermediate in the northern zone. 

### 3.4. Accessibility of ASRH Facilities and Services

With regards to the accessibility of the health facilities and services, the results in Table 4 show that less than half of the facilities in the three senatorial zones were located near the main road, and more than half of the facilities across the zones were situated close to places where adolescents gather in the locality and where schools were located. However, less than half of the facilities in the three zones had opening and closing hours that were convenient for adolescents. Specifically, more than half of the facilities in the three senatorial zones were located in areas accessible to adolescents but less than 50% of the facilities in the three zones had operating hours that were convenient for adolescents. This was however significantly better in the southern zone compared to the northern zone and the central zone.

### 3.5. Appropriateness of ASRH Services 

Overall, the SRH services in the state were not appropriate for adolescents (see Table 5), and all three zones scored poorly. Of note, the PHC in the southern zone had a significantly higher number of peer educator staff and were more likely to provide ASRH services without parental consent.

### 3.6. Quality of ASRH Services

The quality of ASRH services was poor across the three senatorial zones, and no significant differences were found.

## 4. Discussion

One of the most critical ways of meeting the sexual and reproductive health needs of adolescents is through the provision of comprehensive, adolescent-friendly sexual and reproductive health services. Most adolescents are in need of various SRH services and studies in Nigeria and elsewhere have shown they at times self-medicate or go to unqualified providers for help, and may risk their health and lives as a result of unsafe practices [25,29]. The extent of preparedness of health facilities to provide ASRH services is critical to the utilization of these services by adolescents, and ultimately to the reduction and prevention of unintended teenage pregnancies and maternal mortality which are unacceptably high in Nigeria [30], as well as to the reduction of STI and HIV infections among adolescents [21]. The aim of the current study was to assess the provision of ASRH services in health facilities in Plateau State, Nigeria, in light of the national policy that ASRH be integrated into PHC facilities across the country. 

For SRH services to be adolescent friendly, several factors must be considered, according to WHO guidance: the availability of dedicated space and equipment for ASRH, the availability of specific ASRH services, the accessibility of ASRH services, the appropriateness of ASRH services and the quality of ASRH services [26]. The results of the assessment reported in this study showed that, overall, PHCs across the senatorial zones in Plateau State do not meet the minimum requirements for the provision of ASRH services in any of the five domains. The national target of 50% PHCs providing adolescent-friendly SRH services was only met for the accessibility of ASRH in the central and southern senatorial zones, with the northern zone approaching this target. Similar results were seen in a survey conducted in other parts of Nigeria, where health facilities had no dedicated space and equipment for ASRH and the facilities did not provide specific ASRH services [22]. This finding is also in line with results of an assessment of adolescent and youth-friendly health services in PHCs in two provinces in South Africa using a similar approach, which found that none of the primary health care facilities met the guideline standard for providing adolescent and youth friendly services [31]. 

In the present research, the majority of the PHCs across the three zones did not meet basic requirements for the provision of ASRH services. Facilities that were sampled in this study generally lacked space and equipment for ASRH. For example, many of the facilities did not have a dedicated space for consultations with adolescents, no waiting rooms to separate adult clients from adolescents and no dedicated examination rooms for adolescents. The equipment was all adult size, and they lacked sphygmomanometers with the cuff for adolescents and had no adolescent size speculum for vaginal examination. Traditionally, health facilities in Nigeria in general are designed for adults, with no consideration for adolescents. Many of the facilities lack separate spaces for adolescents, have no waiting room to separate adult clients from adolescents and have no separate examination room for adolescents [23,24].

The provision of essential ASRH services was below the benchmark stipulated by the national policy. Very few PHCs in the three zones offered critical ASRH services such as counselling on sexuality, contraception or safe sex/STI prevention. Although family planning services were available in most facilities, these are usually not targeted at adolescents but at married women. Counselling on gender-based violence was least likely to be provided, probably because it is not recognized as a reproductive health problem. This finding is in line with an assessment of youth-friendly health services in Nigeria, which also found that specific ASRH services, such as counselling on safe sex and provision of contraception, were either not available or of poor quality in centres designated to provide these services [25,26]. Only the availability of counselling on HIV testing scored above 50% across the zones. This is not surprising considering the government focus on HIV prevention and treatment, which has motivated many non-governmental organizations to provide services targeted at HIV prevention in most PHC facilities. Research from Uganda and South African similarly showed that the provision of specific ASRH services was limited, with the exception of voluntary counselling and testing (VCT) for HIV that was fairly widely provided [22,24]. Comparisons across the senatorial zones showed that the southern zone scored best with respect to the provision of in counselling for safe sex and post-abortion care, but the reason for this difference could not be established in this study. It may be that because the southern zone is a rural area, this is considered more in need of SRH care services and hence benefited from the support of non-governmental organizations (NGOs) active in the region.

ASRH services should be accessible to those who need it, and findings show that most of the facilities were situated close to schools and other places where adolescents gather in the localities. While this could have been a strength and opportunity to leverage on in reaching out to adolescents, that was not the case, as it was found that the opening and closing hours of the majority of the facilities were not suited to adolescents’ needs. The facilities did not operate at hours convenient for adolescents (e.g., after school), and had no separate hours for adolescent counselling. Again, the Southern senatorial zone did best, but still less than 50% of facilities had convenient operating hours. Other studies have shown that adolescents have preferred hours for visiting health facilities, usually when adults are not around, because of privacy and confidentiality [17,31]. Adolescents usually do not like to discuss their sexual and reproductive health issues if adults can overhear conversations, and a study from Tanzania found that lack of privacy was one of the reasons for poor utilization of ASRH services [17,32].

Health education is an important approach to meeting the SRH needs of adolescents, which was lacking in almost all the PHCs. There were no appropriate SRH health education materials like posters, and peer education was not provided in most of the facilities because of an absence of peer educators. The southern senatorial zone also scored best with respect to ASRH health education with respect to compared to the others. Outreach services have been identified as a promising approach to meet adolescents’ SRH service needs, bearing in mind that they value confidentiality and most will not want their parents to know they access such services [33,34,35]. Unfortunately, not many health facilities across the state were providing outreach services, despite the proximity of the health facilities to adolescents.

It is important that health care providers are knowledgeable about ASRH, and that ASRH service delivery is support by a policy guideline to ensure the availability and quality of services. Unfortunately, policy guidelines were generally lacking in PHCs in the three senatorial zones. Furthermore, very few facilities had staff specifically trained to deliver ASRH, especially in the central zone. A lack of staff training for ASRH was also reported in studies conducted in Tanzania and Nepal, where a lack of qualified staff affected the utilization of ASRH services [33,36,37,38].

### Strength and Limitations

Some limitations of the current research need to be acknowledged: The study could have been subject to a social desirability bias in healthcare workers’ responses, i.e., a desire to present the facility in a good light in order to comply with policy recommendations. However, because the data collectors insisted on viewing and inspecting the infrastructure and the equipment, we feel this bias has been avoided. Additionally, our study was limited to public primary health care facilities and did not explore the private health facilities—it could be possible that some private health facilities may have adequate ASRH services in place. It would have been more holistic to assess all categories of health facilities. However, as the public health facilities are the first go-to for the majority of adolescents, particularly in the rural areas, we felt that ASRH at these facilities should be investigated first.

## 5. Conclusions

This study showed that most of the PHCs that were surveyed in this study lack the basic infrastructure and equipment for the provision of ASRH and also to do provide essential ASRH services. Only few PHC facilities, especially in the southern senatorial zone, had a dedicated counselling area for adolescents and also provided essential ASRH services, such as counselling on safe sex and prevention of pregnancy. While about half of the facilities were situated around schools and other places where adolescents gather, outreach activities to contact and provide SRH services for adolescents were not carried out in most of the facilities. Most of the health care providers had no training in delivering ASRH services, and thus likely lack skills and competencies in ASRH service delivery. This may explain why even basic ASRH services were generally not provided. Overall, the availability of ASRH space and equipment, specific ASRH services and appropriate ASRH service delivery were suboptimal across the three senatorial zones in Plateau State, as was the quality of ASRH services. 

## 6. Recommendation

While the health system is not the only factor affecting the sexual and reproductive health of adolescents in Nigeria, the lack of basic infrastructure and essential SRH services for adolescents will adversely affect any efforts to achieve meaningful reduction in adolescents’ sexual and reproductive health challenges, which in turn contribute to the poor maternal health indices in Nigeria. To effectively promote ASRH, the country needs to take a critical next step of going beyond policy formulation to ensuring the implementation of services to achieve the policy targets. This likely requires a restructuring to make PHCs more adolescent friendly, and additionally requires an urgent strengthening of the training of health care providers to increase skill in ASRH service delivery. These actions should be supported by standard management guidelines and procedures issued at government-level and that need to be made available in all PHCs providing ASRH services in Plateau State, and Nigeria more generally.

## Figures and Tables

**Table 1 ijerph-18-01369-t001:** Proportion of health facilities per senatorial zone meeting all criteria for the specific domains of adolescent friendly sexual and reproductive health services.

Domains of Adolescent Friendly Sexual and Reproductive Health Services	Senatorial Zone			χ^2^	*p*-Value
	Northern	Central	Southern		
Availability of structure for ASRH	7.6%	5.4%	10.1%	1.652	0.44
Availability of services for ASRH	17.8%	13.5%	22.8%	2.218	0.33
Accessibility of ASRH services	45.5%	50%	55.3%	0.808	0.67
Appropriateness of ASRH services	10.2%	6.5%	13%	1.800	0.41
Quality of ASRH services	25.3%	22.6%	28.2%	0.500	0.78
Overall	21.3%	19.6%	25.9%	0.925	0.63

**Table 2 ijerph-18-01369-t002:** Proportion of health facilities per senatorial zone with dedicated adolescent sexual and reproductive health (ASRH) space and equipment.

	Senatorial Zone			χ^2^	*p*-Value
	Northern	Central	Southern		
Space for ASRH	1.5%	1.1%	1.4%	0.071	0.97
Counselling area	7.6%	5.4%	16.9%	6.541	0.04
Examination room	16.7%	5.4%	15.5%	6.261	0.05
Waiting room	4.5%	1.1%	0%	4.623	0.10
Equipment	7.6%	14%	14.1%	1.740	0.40

**Table 3 ijerph-18-01369-t003:** Proportion of health facilities per senatorial zone with dedicated ASRH services.

	Senatorial Zone			χ^2^	*p*-Value
	Northern	Central	Southern		
Counselling on sexuality	12.1%	7.5%	15.5%	2.621	0.27
Counselling on Contraception	10.6%	7.5%	16.9%	3.546	0.17
Counselling on STI	16.7%	14%	26.8%	4.518	0.10
Counselling on safe sex	18.2%	9.7%	25.4%	7.132	0.03
Counselling on GBV	6.1%	6.5%	11.3%	1.654	0.43
Management of GBV	1.5%	2.2%	6%	2.355	0.30
Counselling on VCT	62.1%	55.9%	52.1%	1.709	0.49
Post abortion care	7.6%	2.2%	15.5%	10.642	0.01

**Table 4 ijerph-18-01369-t004:** Proportion of health facilities per zone with ASRH services that are accessible.

	Senatorial Zone			χ^2^	*p*-Value
	Northern	Central	Southern		
* Distance from main road ^#^	40.9%	39.8%	47.9%	1.289	0.55
Distance from where adolescents gather	56.1%	67.7%	63.4%	2.538	0.32
Distance from school to locality	54.5%	67.7%	64.8%	2.710	0.22
* Convenient opening/closing hours	30.3%	24.7%	45.1%	7.724	0.02

* A distance within 30 min’ walk is considered close; ^#^ After school availability is considered convenient opening/closing hours.

**Table 5 ijerph-18-01369-t005:** Proportion of health facilities per zone with appropriate ASRH services.

	Senatorial Zone			χ^2^	*p*-Value
	Northern	Central	Southern		
Specific clinic hours for adolescents	4.5%	5.4%	2.8%	0.645	0.73
Availability of posters Yes	1.5%	3.2%	2.8%	0.443	0.79
Dedicated Peer education staff Available	1.5%	1.1%	9.9%	9.603	0.01
Organized Outreach services Yes	3%	0%	2.8%	2.797	0.25
Services offered without parental consent Yes	47%	29%	45.1%	6.388	0.03
Guideline for ASRH Present	0%	0%	0%	n/a	n/a
Staff trained on ASRH Trained staff	7.6%	5.4%	8.5%	0.655	0.73
Referrals made Yes	68.2%	62.4%	76.1%	3.483	0.18

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
