# Peer review of "Adolescent Sexual and Reproductive Health Care Service Availability and Delivery in Public Health Facilities of Plateau State Nigeria"

_ijerph, 2021, doi:10.3390/ijerph18041369_

Round 1
Reviewer 1 Report
The manuscript entitled " Adolescent Sexual and Reproductive Health Care Service Availability and delivery in Public Health Facilities of Plateau State Nigeria" by Esther Awazzi Envuladu and colleagues investigate the provision and quality of reproductive and sexual health counseling to adolescents in a population of Nigeria's Plateau State. Overall, the results show that sexual health care/counseling to adolescents is not provided and when is available is of very poor quality. I think the results are interesting and can foster governmental policies aimed to improve the quality of health support provided. Some minor issues were identified and listed below.
Page 1, line 12: “To assessed...” change to “To assess…”
Page 1, lines 12 and 14: please define ASRH, WHO and PHC the first time they appear in the abstract section
Page 1, lines 43 and 44: please correct “adoslecents girls…” to “adolescent girls…”
Page 2, line 59: please define WHO the first time it appears in the abstract section.
Page 2, line 60: please correct the abbreviation of Sexual and Reproductive Health as SHR throughout the manuscript.
Page 7, line 207: please correct “the results shows…” to “the results show…”
Page 11, line 304: please define NGO the first time it appears in the manuscript.
Reviewer 2 Report
This is an interesting, well written and useful paper. Evidence for a much needed change in Adolescent Sexual and Reproductive Health service provision in Plateau state Nigeria.
Minor points:
- Line 67: I think the authors should highlight the fact that the policy is now a decade old
- Line 84: ‘the Southern zone could be said to be rural.’ – Is it rural or not? Not sure if this is just a language issue but it doesn’t seem to me like something that would be difficult to categorise?
- Line 107: ‘administered the survey questionnaires to the heads of the facilities’ – how were the questionnaires administered? Did the researchers ask the questions or did the respondents complete on paper?
- Table 4. Presumably opening/closing hours should be ‘convenient’ rather than close?
- The authors should add a limitations section at the end of the discussion
- As the findings of this study could and should be used for informing policy decisions and further research, I would like to see a sub-headed section in the discussion ‘recommendations for policy and practice’ (or similar) that brings together and highlights the information currently provided in the discussion. I think this would make sure the recommendations are easily accessible to those who don’t have time to read the whole paper/discussion.
Some minor grammar mistakes:
- Title: Make the word ‘delivery’ upper case
- Abstract: Spell out acronyms in first instance
- Line 11: first line: Change ‘To assessed’ to ‘ To assess’
- Line 12: Change capital P in Primary to lower case
- Line 21: Should be ‘operating at a convenient time’ or ‘operating at convenient times’
- Lines 30 and 35: Change ‘condom less sex’ to ‘sex without condoms’ or ‘unprotected sex’
- Line 53 change ‘lower’ to ‘low’
- Line 295: Changed ‘gender base violence’ to ‘gender based’
Referencing
There are several instances in which statements are not referenced appropriately:
- Line 40/41: The following statement should be supported with an appropriate reference. ‘In most instances, adolescent pregnancies in Nigeria end in termination, despite the illegality of abortion.’
- Line 46: The following statement should be supported with an appropriate reference ‘In most cases, these girls end up with serious complications while some die in the process, adding to the burden of maternal mortality in the country.
- Line 51: The following statement should be supported with an appropriate reference Furthermore, teenage pregnancies that are not terminated are associated with maternal reproductive risks for the teenage mother, since they are at higher risk of pre-eclampsia and eclampsia, antepartum hemorrhage and feto pelvic disproportion, as well as obstructed labor and its sequelae, notably genital fistulae.
Reviewer 3 Report
Envuladu et al. examined the availability, accessibility, appropriateness, and quality of Adolescent Sexual and Reproductive Health (ASRH) Service in Nigeria's public health facilities. The authors found that most facilities lack ASRH services, which may significantly impact disease control and prevention. The manuscript is well-organized and well-written. However, there are some issues in statistical testing. The manuscript shall be suitable for publication with major revisions.
Specific comments.
- Statistical testing. The authors conducted Chi-square tests to compare proportions of health facilities by different senatorial zones. However, the comparison is subject to potential confounding bias. The statistical testing should account for demographics (age, sex, etc.) and socioeconomic factors in different geographical locations.
- Abstract. Page 1, line 12. Typo. “assessed” should be “assess”.
- Abstract. Page 1, line 12. Write out “ASRH”. Define abbreviations at their first appearance in the manuscript.
- Abstract. Page 1, line 21. Specify “convenient time for adolescents”. It is unclear.
- Abstract. It is important to talk about the public health implications of this study.
- Page 1, lines 37-38. What’s the difference between the “pooled adolescent pregnancy rate” and “pregnancy rate”? Clarify that.
- Page 1, line 42. Is the yearly rate of 25 abortions per 1000 women illegal? If so, clarify that.
- Page 2, line 63-64. Please explain the term “adult-centered, not specific to adolescents’ needs”.
- Page 2, lines 80-81. Please describe “the national reproductive health policy for adolescents in Nigeria”.
- Methods. It would be helpful to describe the characteristics of the general population in Plateau State, Nigeria.
- Methods. Please describe the statistical testing used in the analyses. Fisher’s exact tests are preferred in analyzing a small sample size (N<5) compared to Chi-square tests.
- Discussion. Need to add a discussion regarding public health implications and recommendations for policy change.
Round 2
Reviewer 3 Report
The authors have addressed all my comments.